# Epidemiological characteristics of hand, foot and mouth disease in Pingdu, Shandong, China from 2013 to 2023 and DLNM analysis of its variation with temperature

Hua Zhang[1], ChangLan Yu[2], DaiXia Yang[3], Wei Zhang[1], ShouJie Dai[3]*, Hui Lv[4,5]*, XiaoLin Liu[4,5]*

**1** Dazeshan Town Health Center, Pingdu City, Qingdao, China, **2** Yantai Center for Disease Control and Prevention, Yantai, China, **3** Pingdu Center for Disease Control and Prevention, Qingdao, China, **4** Shandong Center for Disease Control and Prevention, Jinan, China, **5** Shandong Province Key Laboratory of Intelligent Monitoring, Early Warning, and Prevention of Infectious Diseases, Jinan, China

* pdcdccfk006@qd.shandong.com (SD); 272899468@qq.com (HL); sjobshuier@163.com (XL)

## Abstract

### Background

The purpose of this study is to describe the epidemiological situation of HFMD in Pingdu over the past decade, and investigate the relationship between environmental factors, specifically temperature, and the incidence of hand, foot and mouth disease.

### Methods

Statistical techniques, including Distributed Lag Non-linear Models and spatial autocorrelation analysis, were employed to elucidate epidemiological characteristics of hand, foot and mouth disease in Pingdu and the non-linear effects time-lagged relationships of temperature on the incidence.

### Results

The incidence of hand, foot and mouth disease in Pingdu exhibits seasonal distribution, and the incidence rate is highest from May to August each year. The spatial distribution shows almost no spatial autocorrelation. Children under the age of 7 account for 91.09% of HFMD cases, with an obvious trend of increased incidence in older age groups by 2023. Notably, severe cases predominantly occurred in children under 3 years old, and EV-A71 accounts for a higher proportion compared with other enteroviruses. The pathogen types of hand, foot and mouth disease have changed from mainly EV-A71 and CVA16 to other enteroviruses. When the daily maximum temperature reaches 33.4°C, the relative risk (RR = 1.33) is highest at the one lag day.

**Data availability statement:** All data generated or analyzed during this study are included in this published article. The dataset (the daily number of HFMD cases from 2013 to 2023) used and/or analyzed during the current study were obtained from Pingdu Center for Disease Control and Prevention, Qingdao, China. We have no right to make this dataset public. If you need the dataset, you can contact with the organization (Tel: +86-532-88328906; Address: NO.222 Changzhou Road, Pingdu, Qingdao, People's republic of China, 266700).

**Funding:** The author(s) received no specific funding for this work.

**Competing interests:** The authors have declared that no competing interests exist.

## Conclusion

This study reveals the epidemiological characteristics and climate risk factors of hand, foot and mouth disease in Pingdu. It is important to note that children, especially those under the age of 3, are the key population for the prevention and control of hand, foot and mouth disease. It is recommended that health authorities incorporate temperature into the formulation of hand, foot and mouth disease prevention and control policies.

## 1 Introduction

Hand, foot and mouth disease (HFMD) is a prevalent viral illness predominantly affecting children, characterized by vesicular lesions on the oral mucosa and skin of the hands and feet [1,2]. Over the last two decades, HFMD has emerged as a significant public health concern, particularly in the Asia-Pacific region, where outbreaks have led to severe neurological complications and fatalities in young children [3,4]. The rapid transmission of HFMD not only poses substantial health risks to affected individuals but also results in considerable economic burdens on healthcare systems and families [5,6].

Current treatment options for HFMD are largely limited to symptomatic relief, as there are no specific antiviral therapies approved for use against the viruses responsible for this disease [7]. Additionally, while vaccines targeting EV-A71 have been developed and implemented in some regions, there remains a significant gap in protective measures against other prevalent serotypes, including CVA6 and CVA10, which have been associated with recent outbreaks [8,9]. This highlights the urgent need for comprehensive prevention strategies and effective public health interventions to mitigate the impact of HFMD outbreaks [10].

The relationship between environmental factors and the incidence of HFMD has been a focus of recent researches [11–15]. Studies have demonstrated varying correlations between environmental parameters and the incidence of HFMD, suggesting that temperature may play a crucial role in triggering outbreaks. Understanding these associations is essential for developing effective public health strategies and implementing timely interventions during peak seasons of HFMD occurrence. This study employs Distributed Lag Non-linear Models (DLNM) and spatial autocorrelation analysis, to explore the influence of temperature variations and spatial factors on HFMD incidence in Pingdu. These methodologies provided insights into the dynamics of disease transmission in relation to environmental factors.

Pingdu is a county-level city with relatively consistent temperatures across its territory, making it an excellent choice for studying the impact of temperature changes on HFMD incidence. By integrating demographic data, environmental factors, and advanced statistical techniques, this study aims to fill existing research gaps regarding HFMD dynamics and contribute to a more nuanced understanding of the disease's epidemiology. The findings are expected to enhance the ability of public health officials to implement targeted interventions and improve awareness among vulnerable populations, ultimately reducing the burden of HFMD on children.

## 2 Data and methods

### 2.1 Geography and climate

Pingdu locates in the Northeastern China and ranges in latitude from 36°28' to 37°02'N and in longitude from 119°31' to 120°19'E. It bounds on the northeast by Yantai, on the north and the southeast by Laixi (Qingdao), on the west and the south by Weifang. Its land area is approximately 3,176 km². At the end of 2023, the permanent resident population of Pingdu was approximately 1.18 million.

Pingdu experiences a humid continental climate with four distinct seasons, characterized by hot, humid summers and cold, dry winters (with occasional snowfall). The annual average maximum daily temperature is approximately 21°C, with monthly average max temperature ranging from 2°C in January to 31°C in July. The city receives an average annual precipitation of 710 mm, predominantly concentrated in summer under the influence of the East Asian monsoon, with July typically recording the highest rainfall and relative humidity.

### 2.2 Data collection

The daily cases data of HFMD in Pingdu from January 2013 to December 2023 were obtained from Notifiable Disease Surveillance System (NDSS) for Disease Control and Prevention. The diagnostic criteria of HFMD were based the clinical criteria published by the National Health Commission of the People's Republic of China [16]. Population data for Pingdu were obtained from the Pingdu Bureau of Statistics "Pingdu Yearbook". Daily m temperature data were collected from NOAA website [17].

### 2.3 Spatial autocorrelation

Spatial autocorrelation refers to potential interdependence between the observed data of certain variables within the same distribution area, including global and local autocorrelations. Global autocorrelation reflects the overall spatial aggregation of a certain phenomenon in a regional unit, and local spatial autocorrelation analysis reflected the spatial relationships of different element indicators in local areas. This study uses Moran's I statistic as an indicator and analyzes it at the town level. All analyses were conducted using ArcGIS (version 10.8.2, USA).

### 2.4 Distribution-lagged nonlinear model (DLNM)

A distributed lag non-linear model (DLNM) with quasi-Poisson regression was applied to evaluate the association between daily highest temperature and incidence of HFMD in Pingdu. The DLNM model was defined by the following formula [18]:

$$\log[E(Y_t)] = \alpha + cb(Tmean_t, lag = 14) + ns(time_i, df = 7/year) + \beta\, DOW_t$$

where: Yt is the number of cases on day t, $\alpha$ is the model intercept. $Tmean_t$ is the daily max temperature on day t. cb refers to the crossbasis function, which specifies the exposure-lag-response relationship simultaneously in the exposure-response and lag response dimensions. A natural cubic splines (ns) with 4 degrees of freedom was used for the lag-response relationship. The lag day up to 14 days reflcetes the maximum lag day of the temperature, $time_t$ is the long-term trend, df is the degrees of freedom, $\beta$ is the coefficient, and $DOW_t$ is the day-of-the-week effect.

### 2.5 Data analysis methods

Data cleaning and statistical analysis were performed using Excel 2019 and R 4.5.1 software, with P < 0.05 indicating statistical significance. Time series analysis and incidence effect plots in this study were generated using the "dlnm, ggplot2, splines, viridis" packages in R 4.5.1 software.

# 3 Results

## 3.1 Time distribution

From 2013 to 2023, a total of 3,924 HFMD cases were reported in Pingdu, with an annual incidence rate of 29.02 per 100,000. The incidence rate decreased significantly in 2016, with a slight increase from 2017 to 2019. In 2020, the incidence rate was extremely low due to COVID-19 lockdown measures, and it has been rising annually since the pandemic, returning to the annual average level in 2023 (Fig 1).

HFMD disease incidence in Pingdu exhibit a certain seasonal distribution, with May to August being the peak season for HFMD. The epidemic begins to rise rapidly in May, peaks in July, and cases from May to August accounted for 68.6% of all the total. From 2020 to 2022, the incidence rate decreased due to COVID-19 lockdown measures. After the end of 2022, the incidence rate began to rise again, reaching a peak in July and August 2023. This represents a delay in the peak period compared to previous years, which typically occurred in May and June (Fig 1).

## 3.2 Spatial distribution

From 2013 to 2023, all 17 towns in the city reported HFMD cases. Spatial autocorrelation analysis of revealed no significant association (Moran's I = 0.075, Z = 0.762, P = 0.446). Annual spatial autocorrelation analysis indicated that spatial autocorrelation was only present in 2022 (Moran's I = 0.250, Z = 2.060, P = 0.039), with two towns showed low-high clustering, while three towns exhibited high-high clustering (Table 1).

## 3.3 Population distribution

From 2013 to 2023, among the reported HFMD cases in Pingdu, the age distribution was concentrated in children under 7 years old, accounting for 91.90% of the total. The highest incidence rate was observed in the 1~2 years old age group, accounting for 31.63%, followed by the 2~3 years old age group for 18.50%, and the 3~4 years old age group for 16.79%. There is a trend of increasing incidence with age in recent years. Especially in 2023, the number of cases in the 6~12 age group and those aged 18 and above showed a significant increase. The number of male cases exceeded that of female ones, with a male-to-female ratio of 1.5:1(Fig 2).

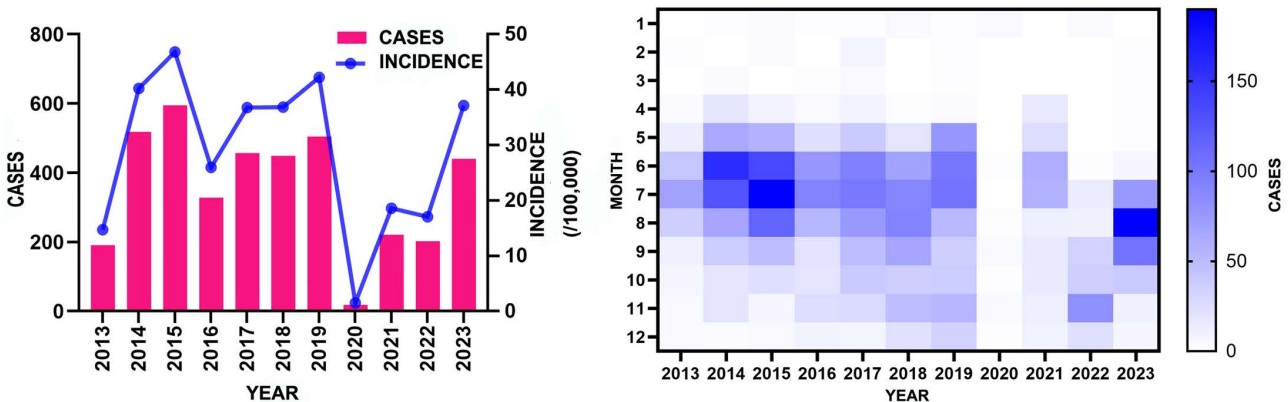

**Fig 1. Time distribution of HFMD cases in Pingdu from 2013 to 2023.**

**Table 1. Spatial autocorrelation analysis of HFMD in Pingdu from 2013 to 2023.**

| Year | Moran's I | Z | P | Spatial autocorrelation |
|---|---|---|---|---|
| 2013-2023 | 0.075 | 0.762 | 0.446 | NO |
| 2013 | −0.175 | −1.126 | 0.260 | NO |
| 2014 | −0.120 | −0.364 | 0.716 | NO |
| 2015 | −0.006 | 0.324 | 0.746 | NO |
| 2016 | −0.164 | −0.656 | 0.512 | NO |
| 2017 | 0.025 | 0.628 | 0.530 | NO |
| 2018 | −0.108 | −0.278 | 0.781 | NO |
| 2019 | −0.083 | −0.122 | 0.903 | NO |
| 2020 | 0.064 | 0.900 | 0.368 | NO |
| 2021 | 0.031 | 0.662 | 0.508 | NO |
| 2022 | 0.250 | 2.060 | 0.039 | YES |
| 2023 | 0.094 | 1.405 | 0.160 | NO |

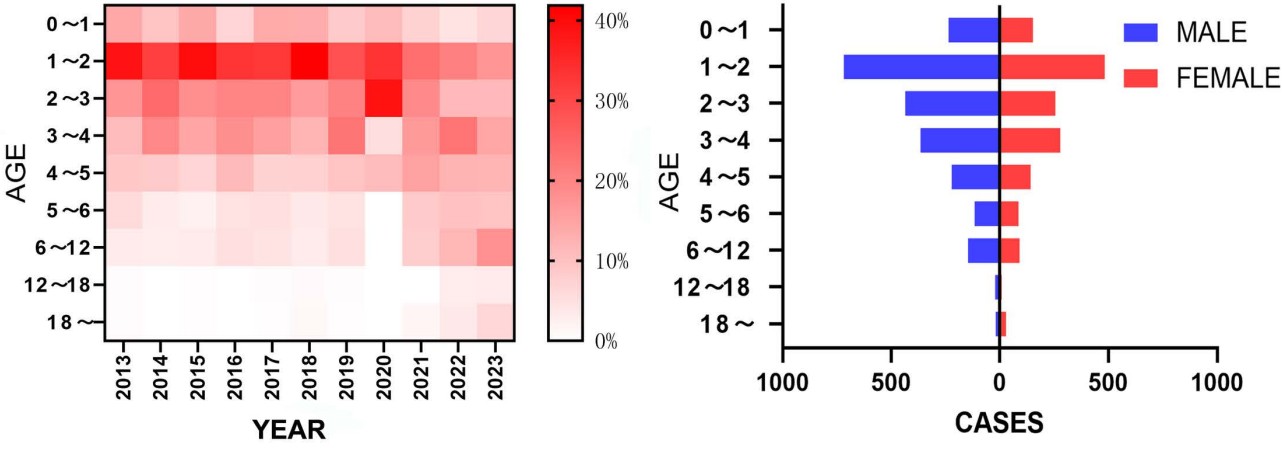

**Fig 2. Age Distribution of HFMD cases in Pingdu from 2013 to 2023.**

### 3.4 Epidemiological characteristics of severe HFMD cases

From 2013 to 2023, a total of 102 severe HFMD cases were reported in Pingdu, with the highest number of severe cases occurring in 2014, accounting for 74.51%. No severe cases were reported after 2018. The results showed that factors such as scattered children ($x^2 = 18.343$, $P < 0.001$), children under 3 years old ($x^2 = 9.978$, $P = 0.002$), occurring between May and August ($x^2 = 9.98$, $P < 0.001$), infected EV-A71 ($x^2 = 401$, $P < 0.001$) are risk factors for severe HFMD (Table 2).

### 3.5 Pathogen serotype composition

From 2013 to 2023, there were a total of 401 laboratory-confirmed cases, including 97 cases of CVA16, 52 cases of EV-A71, and 252 cases of other enteroviruses. From 2013 to 2014, the proportions of EV-A71, CVA16, and other enteroviruses were relatively balanced. Since 2015, other enteroviruses became the main infection types (with CVA16 being the predominant infection type in 2019) (Fig 3).

**Table 2. Distribution of severe and mild cases of HFMD in Pingdu from 2013 to 2023.**

| Risku factors | | Severe HFMD n (%) | Mild HFMD n (%) | $\chi^2$ | P |
|---|---|---|---|---|---|
| Categorization of people | | | | 18.343 | <0.001 |
| | Scattered children | 91 (89.22) | 2645 (69.53) | | |
| | preschool children | 10 (9.8) | 948 (24.92) | | |
| | Student | 1 (0.98) | 167 (4.39) | | |
| Age(year) | | | | 9.978 | 0.0002 |
| | 0~3 | 77 (75.49) | 2282 (59.99) | | |
| | 3~6 | 22 (21.57) | 1207 (31.73) | | |
| | 6~12 | 3 (2.94) | 110 (2.89) | | |
| Month | | | | 9.98 | <0.001 |
| | 5-8 | 91 (89.22) | 2600 (68.35) | | |
| | Other | 11 (10.78) | 1204 (31.65) | | |
| Sex | | | | 1.76 | 0.185 |
| | Male | 67 (65.6) | 710 (58.9) | | |
| | Female | 35 (34.4) | 494 (41.1) | | |
| Pathogen | | | | 401 | <0.001 |
| | CVA16 | 10 (24.39) | 87 (24.17) | | |
| | EV-A71 | 13 (31.71) | 39 (10.83) | | |
| | Other enterovirus | 18 (43.9) | 234 (65) | | |

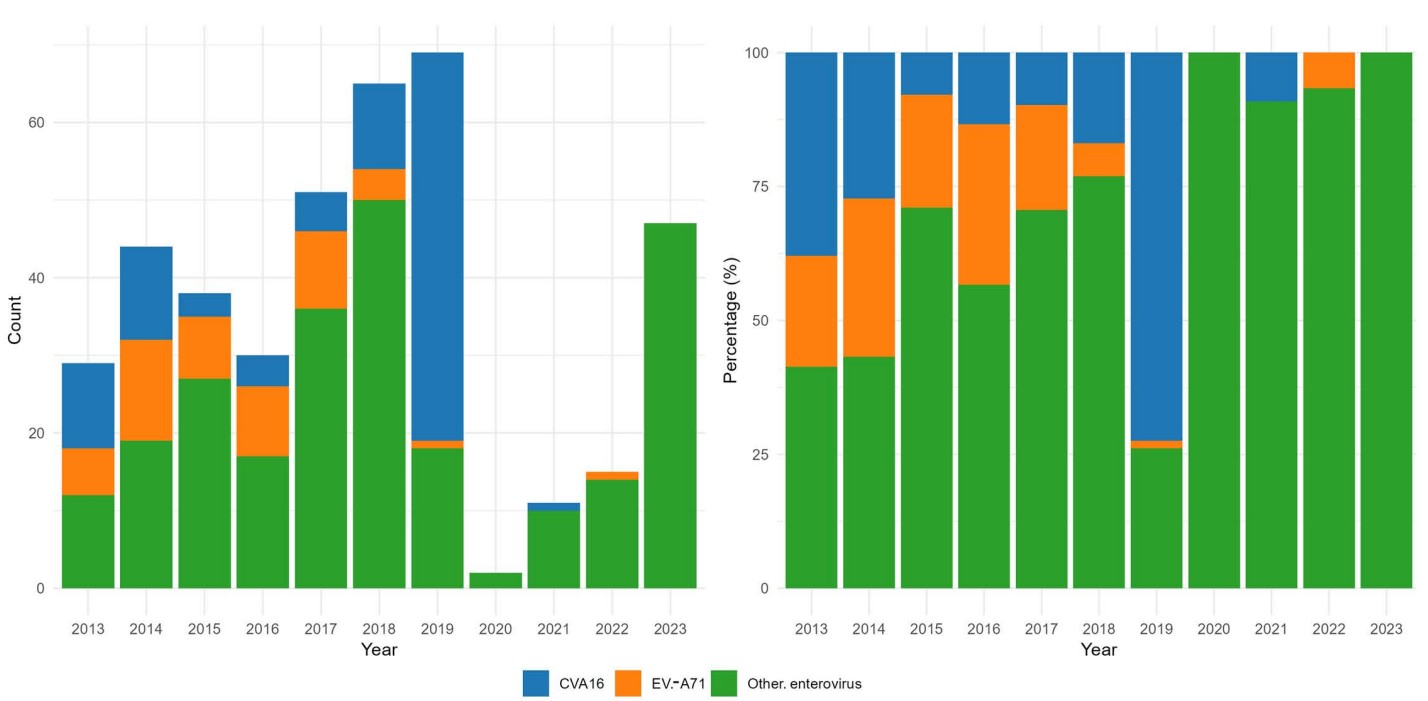

**Fig 3. Distribution of pathogens caused HFMD in Pingdu from 2013 to 2023.**

### 3.6 The impact of daily maximum temperature on the number of HFMD cases

The temperature in Pingdu shows obvious seasonal changes, and the trend in the number of cases is consistent with the trend in temperature changes from 2013 to 2019(due to the impact of COVID-19 pandemic, the data from 2020−2023 was not included in the study) (Fig 4). A three-dimensional plot was created to illustrate the association between daily maximum temperature at different lag days and the risk of HFMD incidence in Pingdu. The results showed that the daily average temperature at different lag days exhibited a nonlinear relationship with HFMD cases, with the relative risk varying with lag time. The relative risk value is highest when the daily maximum temperature is 33.4°C with a lag of 0 days, with RR of 1.33 (95% CI: 1.06, 1.66) vs. RR at mean maximum temperature 21°C. Additionally, when the daily maximum temperature lagged by one day at 1.9°C, the RR was 1.43 (95% CI: 0.64, 3.19) (Fig 5). The cumulative relative risk(CRR) analysis shows that as the temperature rises, the CRR of HFMD increase, with the CRR reaching its highest value at the maximum temperature of 38.7°C. When the predicted temperature is −10°C, the RR fluctuation range is the largest (Fig 6).

## 4 Discussion

Pingdu is located in the inland region of the Jiaodong Peninsula, which is a county-level city under the jurisdiction of Qingdao without a coastline. Its climate combines both coastal and inland characteristics. Research on the epidemiological characteristics of HFMD in Pingdu and its correlation with temperature holds reference value for other areas. Especially

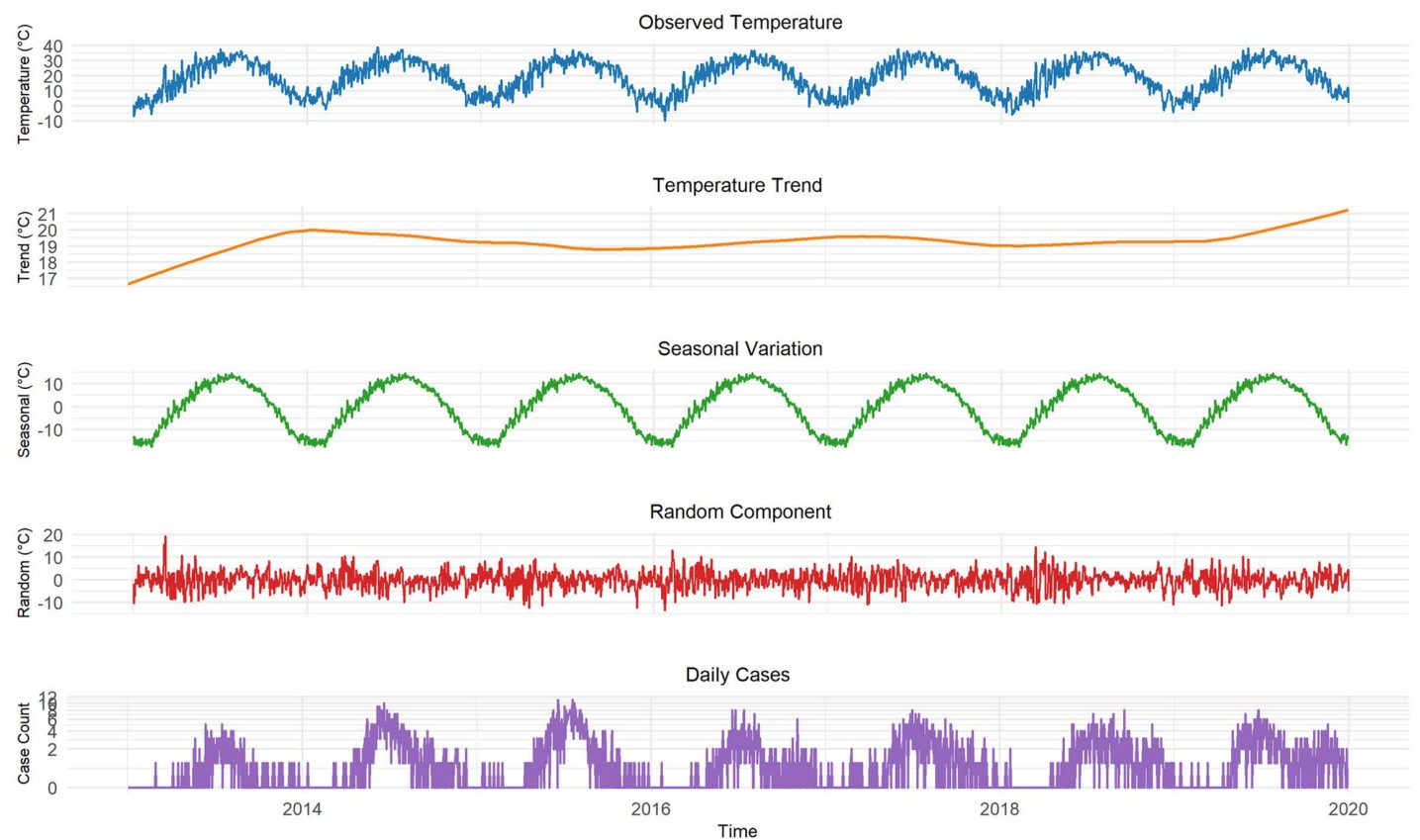

**Fig 4. Distribution of temperature and number of cases in Pingdu from 2013 to 2019.**

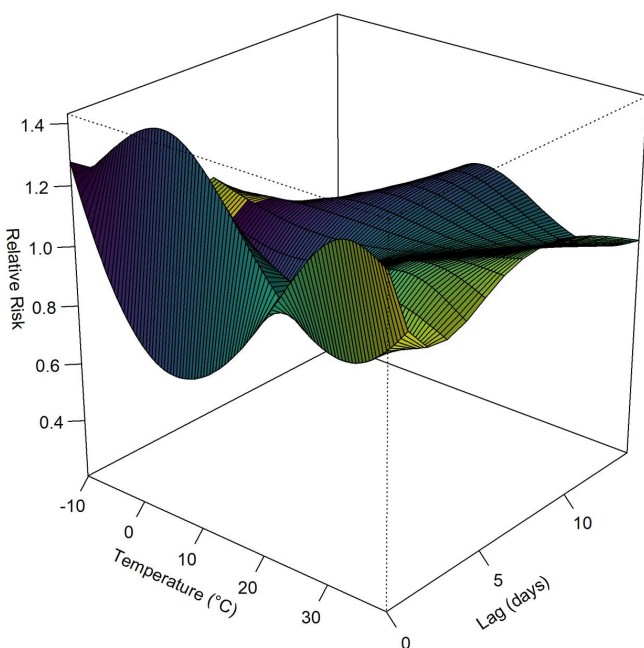

**Fig 5. Lag analysis of the distribution of HFMD relatives risk in Pingdu according to temperature changes.**

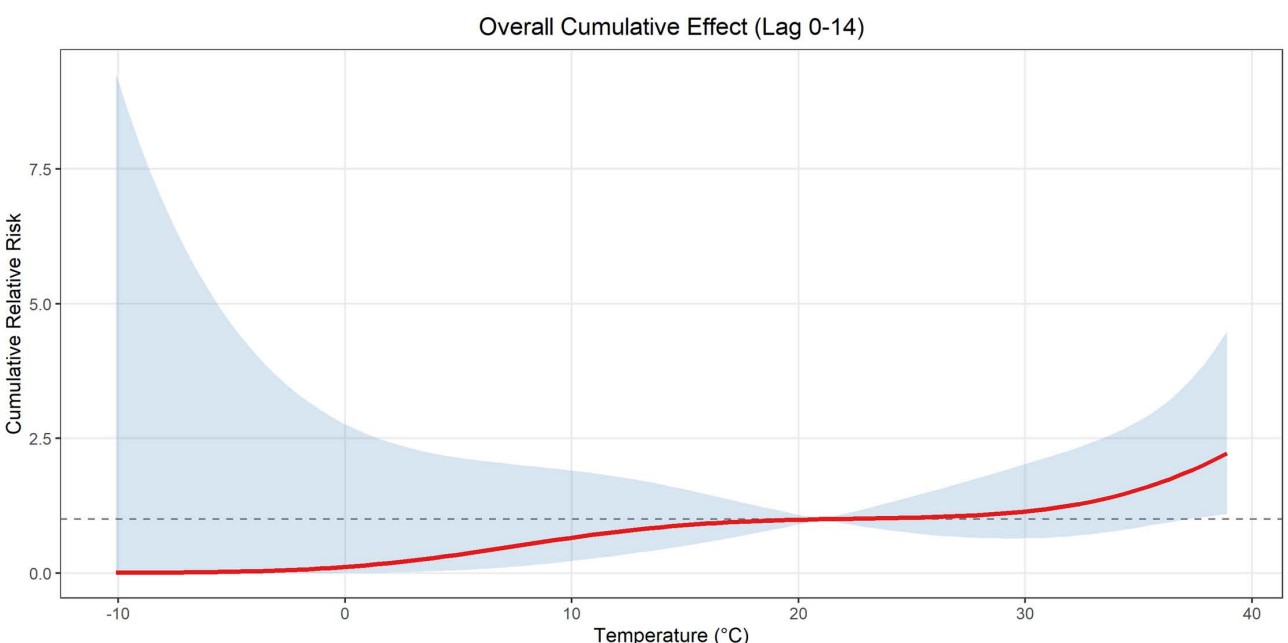

**Fig 6. Cumulative effect of HFMD relatives risk in Pingdu according to temperature changes.**

as a county-level city, Pingdu can provide more accurate analysis of temperature and HFMD incidence compared to large cities.

Prior to the COVID-19 pandemic, the annual incidence rate of HFMD in Pingdu was 33.21 per 100,000 from 2013 to 2019, while the average incidence rate was 29.02 per 100,000 from 2013 to 2023, which was lower than the average incidence rate (145.72 per 100,000) [19] in the national level and that in Shandong Province (84.49 per 100,000) [5]. HFMD incidence in Pingdu exhibits a distinct seasonal distribution, characterized by a single peak, consistent with the single-peak distribution observed in the northern regions [20], but differs from the results of northwestern regions such as Xinjiang [21], and southern regions like Guangzhou [22], which show a double-peak distribution. The difference is associated with differences in temperature, air pressure, relative humidity, precipitation, wind speed, and sunshine duration across regions. In 2020, the incidence rate of HFMD was extremely low, primarily due to non-pharmaceutical intervention measures implemented during the COVID-19 pandemic, including home quarantine, school and childcare facility closures, reduced population gatherings, and low reporting rates [23]. From 2021 to 2023, the incidence rate of HFMD continued to rise, primarily due to the gradual resumption of work and school, followed by prolonged home quarantine and the relaxation of the two-child policy, which may have led to an accumulation and increase in the number of susceptible individuals. Notably, the median age of onset increased annually from 2021 to 2023, with a rise in incidence among school-aged children over 6 years old. This may be of that young children in childcare facilities during the pandemic did not develop immunity due to home quarantine, leading to an accumulation of susceptible individuals. After the pandemic ended, these children entered elementary school and developed the disease, leading to the results that the proportion of HFMD patients over 6 years old showed an upward trend.

Severe cases of hand, foot and mouth disease often present with neurological symptoms and may even result in sequelae [24]. A comparison of the epidemiological characteristics of severe cases with those of mild cases from this survey revealed that children under the age of 3 accounted for a high proportion of cases, with males outnumbering females. In addition, EV-A71 infection and occurred during the period from May to August is also a risk factor for severe disease. Research found that EV-A71 infection increases exosomes in the serum of patients with severe HFMD, which can promote viral replication or promoting cytokine storm [25,26]. It is essential to strengthen health education for children under three years, as well as to promote vaccination. The inactivated EV-A71 vaccine is an economical and effective control measure [27].From 2016 to 2017, three inactivated EV71 vaccines were launched in China, manufactured by Institute of Medical Biology, Chinese Academy of Medical Sciences, SINOVAC and wuhan institute of virology respectively. In this study, the proportion of cases infected by EV-A71 in Pingdu decreased significantly after 2017, and the severity rate decreased to zero. Therefore, the EV-A71 vaccine is effective in preventing severe cases caused by the EV-A71 virus [28], but the overall incidence of HFMD in Pingdu has not changed significantly, reflecting that no immune barrier has yet been established in the population, which is consistent with the findings of Shanghai [29]. The proportion of other enteroviruses (including CVA6 and CVA10) in the pathogen spectrum of HFMD has increased, gradually replacing EV-A71 and CVA16 as the main pathogens causing HFMD, which is consistent with studies in other areas of China and Japan [30,31].

The innovative aspect of this study lies in its comprehensive examination of the interplay between temperature and the incidence of HFMD in Pingdu over a decade. DLNM analysis revealed that HFMD in Pingdu exhibited high relative risk at 33.4℃ with lag time of 0 day. The hot and humid climate in summer and autumn significantly prolongs the survival time of viruses in the external environment, such as on object surfaces and in sewage, increasing the chances of contact transmission. This suggests that high temperature in Pingdu is high-risk factors for the onset of HFMD, and that the incubation period is relatively short. When temperature reaches −1.9°C, the RR appears to rise with lag time of 1 day, suggesting that low temperatures also have a certain impact on the incidence of HFMD. Cumulative effect analysis found that at −10°C, the CRR fluctuation range was the largest, suggesting that extreme weather may be a risk factor for HFMD. However, temperature of −10°C are extremely rare in Pingdu. The results from Qingdao showed that the peak RR concurring

at 30.5°C [32]. This study took into account the large difference between day and night temperatures, so the highest temperature was used as an indicator, and the results were consistent with Qingdao. The result was also aligned with the results from Japan at the same latitude [33,34], while different from Guangzhou at southern regions [35].Another study in Shandong Province (where Pingdu is located) found that the incidence of severe fever with thrombocytopenia syndrome is also higher when the temperature is higher, especially in summer and autumn [36]. Therefore, it is necessary to apply the DLNM model separately to different regions to accurately predict temperature effects [37]. Our findings reveal a significant non-linear relationship between temperature and HFMD incidence, thereby filling critical gaps in the current understanding of the disease's environmental determinants, ultimately informing effective public health strategies and interventions to mitigate the disease's impact on vulnerable populations [38]. However, current prevention and control measures rarely take temperature-related factors into account.

In conclusion, this study highlights significant variations in HFMD incidence over the past decade, revealing crucial demographic factors associated with disease severity. The findings underscore the necessity for continuous surveillance and the implementation of targeted public health interventions, particularly for vulnerable populations such as young children. By elucidating the relationship between environmental factors and HFMD outbreaks, the study found that high temperature is a high-risk factor for the incidence of HFMD in Pingdu City, and prevention and control measures should also be strengthened when the temperature is extremely low. It is important to note that children, especially those under the age of 3, are the key population for the prevention and control of HFMD. It is recommended that health authorities incorporate temperature into the formulation of HFMD prevention and control policies.

## Supporting information

**S1 File. case-temp.**
(CSV)

**S2 File. genotype.**
(CSV)

**S3 File. town-incidence.**
(XLSX)

## Acknowledgments

We would like to acknowledge editors and reviewers for their work and comments.

## Author contributions

**Conceptualization:** Xiaolin Liu.

**Data curation:** ChangLan Yu, DaiXia Yang, ShouJie Dai.

**Investigation:** DaiXia Yang.

**Methodology:** Hui Lv, Xiaolin Liu.

**Resources:** Wei Zhang, ShouJie Dai.

**Software:** ChangLan Yu.

**Supervision:** Wei Zhang, Xiaolin Liu.

**Writing – original draft:** Hua Zhang.

**Writing – review & editing:** Hui Lv.

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
