## [Decision Letter · Decision Letter 0]

2 Sep 2025

Dear Dr. Liu,

Thank you for submitting your manuscript to PLOS ONE. After careful consideration, we feel that it has merit but does not fully meet PLOS ONE’s publication criteria as it currently stands. Therefore, we invite you to submit a revised version of the manuscript that addresses the points raised during the review process.

We look forward to receiving your revised manuscript.

Kind regards,

Ravinder Kumar, PhD

Academic Editor

PLOS ONE

Journal Requirements:

3. Please ensure that you refer to Figure 3, 4, 5 and 6 in your text as, if accepted, production will need this reference to link the reader to the figure.

Reviewers' comments:

Reviewer's Responses to Questions

**Comments to the Author**

1. Is the manuscript technically sound, and do the data support the conclusions?

Reviewer #1: Partly

Reviewer #2: Yes

2. Has the statistical analysis been performed appropriately and rigorously?

Reviewer #1: Yes

Reviewer #2: Yes

3. Have the authors made all data underlying the findings in their manuscript fully available?

Reviewer #1: Yes

Reviewer #2: Yes

4. Is the manuscript presented in an intelligible fashion and written in standard English?

Reviewer #1: Yes

Reviewer #2: Yes

Reviewer #1: Article is good for short communication.

1. I did not find the article conclusive ( while in discussion section some informations are interesting and conclusive).

2. Statistical analysis is included in the article.

3. Allowed data included (author mentioned the limitation).

4. Article is presentable.

Reviewer #2: Manuscript titled "Epidemiological characteristics of hand, foot and mouth disease in Pingdu , Shandong, China from 2013 to 2023 and DLNM analysis of its variation with temperature" by Zhang et al, is a nice piece of work. Through this manuscript author tried to understand relationship between temperature and HFMD. I think this will be important in devising policy in controlling the infection. Although manuscript looks OK in its current form, my comments and suggestions to authors are given below

1) Draft need minor writing improvement

2) Data shown in table can be plotted in histograms

3) Please include vaccines currently in use against the disease (include trade name, vendor, type of vaccine)

4) If possible include loopholes in present disease control measure

5) Mention whether observed correlation of HFMD with temp is also seen with other viral infection in region.

6) Will be nice to explain how rise in temp lead to rise in infection if possible

**Do you want your identity to be public for this peer review?** For information about this choice, including consent withdrawal, please see our Privacy Policy

Reviewer #1: No

Reviewer #2: No

---

## [Author Response · Author response to Decision Letter 1]

7 Sep 2025

Dear editor,

On behalf of my co-authors, we thank you very much for giving us an opportunity to revise our manuscript, we appreciate editor and reviewers very much for their positive and constructive comments and suggestions on our manuscript entitled “Epidemiological characteristics of hand, foot and mouth disease in Pingdu, Shandong, China from 2013 to 2023 and DLNM analysis of its variation with temperature”. (Manuscript Number: PONE-D-25-42952R1).

We have studied reviewer’s comments carefully and have made revision in the revision mode. We have tried our best to revise our manuscript according to the comments. Attached please find the revised version, which we would like to submit for your kind consideration.

We would like to express our great appreciation to you and reviewers for comments on our paper. Looking forward to hearing from you.

Thank you and best regards.

Yours sincerely,

Hua Zhang

Corresponding author: Xiaolin Liu

E-mail: sjobshuier@163.com

---

## [Editor Report · Decision Letter 1]

10 Sep 2025

Epidemiological characteristics of hand, foot and mouth disease in Pingdu , Shandong, China from 2013 to 2023 and DLNM analysis of its variation with temperature

PONE-D-25-42952R1

Dear Dr. Xiaolin Liu

We’re pleased to inform you that your manuscript has been judged scientifically suitable for publication and will be formally accepted for publication once it meets all outstanding technical requirements.

Kind regards,

Ravinder Kumar, PhD

Academic Editor

PLOS ONE
---

## [Editor Report · Acceptance letter]

PONE-D-25-42952R1

PLOS ONE

Dear Dr. Liu,

I'm pleased to inform you that your manuscript has been deemed suitable for publication in PLOS ONE. Congratulations! Your manuscript is now being handed over to our production team.

Kind regards,

on behalf of

Dr. Ravinder Kumar

Academic Editor

PLOS ONE